# Valorization of Phenolic and Carotenoid Compounds of *Sechium edule* (Jacq. Swartz) Leaves: Comparison between Conventional, Ultrasound- and Microwave-Assisted Extraction Approaches

**DOI:** 10.3390/molecules27217193

**Published:** 2022-10-24

**Authors:** Elsa F. Vieira, Suene Souza, Manuela M. Moreira, Rebeca Cruz, Aline Boatto da Silva, Susana Casal, Cristina Delerue-Matos

**Affiliations:** 1REQUIMTE/LAQV, Polytechnic of Porto—School of Engineering (ISEP/IPP), Rua Doctor António Bernardino de Almeida, 4249-015 Porto, Portugal; 2Faculty of Nutrition and Food Sciences, Rua do Campo Alegre, 4150-180 Porto, Portugal; 3REQUIMTE/LAQV, Laboratório de Bromatologia e Hidrologia, Faculdade de Farmácia da Universidade do Porto, Rua de Jorge Viterbo Ferreira 228, 4050-313 Porto, Portugal

**Keywords:** chayote leaves, ultrasound-assisted extraction, optimization, carotenoids, phenolic compounds

## Abstract

Chayote leaves are known for culinary and traditional medicine applications. This work intended to recover carotenoids and phenolic compounds from chayote leaves using the ultrasound-assisted extraction (UAE). A Box–Behnken design was employed to investigate the impact of extraction time, temperature, and ultrasonic power on the recovery of total carotenoids, total phenolic compounds, and antioxidant activities. For comparative purposes, chayote leaf extracts were prepared by maceration (ME) and microwave-assisted extraction (MAE), using the same time and temperature conditions optimized by UAE. Extraction at 50 °C and 170 Watts for 30 min provided the optimal UAE conditions. UAE showed better extraction efficacy than ME and MAE. The HPLC analysis of the extracts showed that the xanthophyll class was the main class of carotenoids, which constituted 42–85% of the total carotenoid content, followed by β-carotene and tocopherol. Moreover, 26 compounds, classified as phenolic acids, flavonols, flavonoids and other polar compounds, were identified in the chayote leaf extracts. Flavonols accounted for 55% of the total compounds quantified (the major compound was myricetin) and phenolic acids represented around 35%, mostly represented by ferulic acid, chlorogenic acid and (+)-catechin. This study revealed the potential of UAE as an effective green extraction technique to recover bioactive compounds from chayote leaves, for food, and for pharmaceutical and cosmetic applications.

## 1. Introduction

*Sechium edule* (Jacq.) Swartz, commonly known as chayote or mirliton, is an edible plant species belonging to the Cucurbitaceae family, along with bitter apple, gourd, cucumber, melon, and pumpkin [1]. This vegetable is widely cultivated in Mexico, Costa Rica, Brazil, and the Dominican Republic [2] and exported to the European Union, United States and Canada, where it assumes fourth place in the most consumed imported products [3]. Although the mature fruit is the most consumed part of the plant, the young leaves, shoots, and the tuberous roots can also be eaten, providing an important source of nutrients [1,4]. For instance, in countries like Malaysia, Taiwan, Thailand, and Vietnam, the consumption of chayote shoots has increased in recent years; the young leaves and tendrils are eaten raw as salad, cooked, or fried [4]. The mature chayote leaves are usually poorly consumed, and mostly used as compost in situ or processing waste after fruit harvest.

Literature reports that tender leaves of chayote contain a considerable amount of protein (2.6–4.8 g/100 g dry weight (DW)), pectin (0.4 g/100 g DW), lipids (0.4–2.3 g/100 g DW), vitamin C (4.6 mg/100 g DW), fiber (12.1 g/100 g DW) [1] and carotenoids, such as lutein (7.4 mg/100 g fresh weight, FW) and β-carotene (4.4 mg/100 g FW) [5]. Chayote leaves are also rich in polyphenols and flavonoids, including C-glycosyl and O-glycosyl flavones [6]; myricitrin (7.5–10.1 mg/100 g DW) and morin (1.9–4.0 mg/100 g DW) [7]. The phytochemical composition of chayote leaves has been associated with promising health properties, due to their antioxidant, anti-inflammatory, and anti-ulcer activities, as well as their hepatoprotective and diuretic properties [8]. For instance, infusions of the leaves are used to dissolve kidney stones and to assist in the treatment of arteriosclerosis and hypertension [1]. Hence, the recovery and utilization of valuable compounds from chayote leaves could represent an important solution in waste management, and their antioxidant constituent extraction would be advantageous for potential use in food, and for the pharmaceutical or cosmetic industries. For this purpose, an effective and sustainable method to apply to the extraction of chayote leaves is necessary. To the best of our knowledge, only four studies (Table 1) have reported on the extraction of carotenoids and phenolic compounds from chayote leaves, and little research has been done to investigate the influence of extraction conditions on the extraction yield, or to investigate the application of modeling techniques to precisely determine the optimal conditions to achieve the maximum carotenoid and phenolic yields from this plant material.

Conventional extraction (maceration) has been the popular method for recovering bioactive compounds from chayote leaves [4,7,8], Table 1. Chao et al. (2014) [7] compared the nutritional and phytochemical profiles and the in vitro antioxidant activity of leaf extracts of green and yellow chayote varieties. Extracts were prepared at 75 °C for 5 h with acidified methanol and the authors reported a higher total phenolic content (TPC) of 2.62 mg gallic acid equivalents (GAE)/g DW for the leaf extract of green chayote; the yellow chayote variety presented a TPC value of 0.63 mg GAE/g DW. The chromatographic analysis showed that myricetin and morin were the principal antioxidant constituents in both leaf extracts, while kaempferol and quercetin were only detected in the leaf extracts from the yellow chayote variety [7]. Loizzo et al. (2016) [8] prepared methanolic extracts of chayote leaves by maceration; the mean values of TPC were 89.3 mg chlorogenic acid equivalents/g DW, and the total carotenoid content (TC) was 0.05 mg/g DW [8]. More recently, Chang et al. (2021) [4] compared the efficacy of four solvents (hexane, ethyl acetate, methanol, and water) on phytochemical extraction from chayote leaves. Extracts were prepared by maceration (2 h at room temperature) and the authors reported that water extract had the highest extraction yield (24.04%) and TPC (5.75 mg GAE/g DW), while the methanolic extract presented the highest total flavonoid content (5.02 mg Quercetin equivalents/g DW) and β-carotene content (0.16 mg/g DW) [8]. Ultrasound-assisted extraction (UAE) is considered one of the most practical extraction techniques for recovering carotenoids and phenolic compounds from plant sources, because of its high efficiency and the popularity of the ultrasonic equipment [10,11,12,13]. Furthermore, unlike conventional extraction, UAE allows the use of low temperature and conservation of heat-sensitive compounds [14]. Although there are well-established advantages of this green-extraction technique, the application of UAE in chayote leaves has been limited. Only Kim et al. [9] used an ultraturrax as extraction equipment and 70% ethanol as extraction solvent, but the authors did not optimize the UAE conditions, namely temperature, time, and ultrasounds amplitude. The extraction yield was 10.7% and the TPC reported was 26.5 mg GAE/g DW [9]. Considering these achievements, this work aimed to apply response surface methodology (RSM), using the Box-Behnken design, to investigate the effects of extraction time, extraction temperature and ultrasonic power on the simultaneous recovery of carotenoids, phenolics, and antioxidant capacity from chayote leaves. For comparative purposes, chayote leaf extracts were prepared by maceration (ME) and microwave-assisted extraction (MAE), using the same time and temperature conditions, optimized by UAE. The carotenoid and phenolic profiles of the UAE, ME and MAE extracts were characterized by high-performance liquid chromatography techniques. The findings from this study are important to extend knowledge on the selection of the best extraction method of carotenoids, phenolics and antioxidant compounds from chayote leaves, to obtain extracts with interest for nutraceutical, pharmaceutical and cosmetic applications.

## 2. Material and Methods

### 2.1. Chemicals

Tocopherols (α-, β-, γ-, and δ-) and tocotrienols (α-, β-, γ-, and δ-) were acquired from Supelco (Bellefonte PA, USA,) and Larodan AB (Malmö, Sweden). Retinyl palmitate and retinol were obtained from Fluka (Buchs, Switzerland) and pheophytin was acquired from DHI (Hørsholm, Denmark). The 2-Methyl-2-(4,8,12-trimethyltridecyl)-chroman-6-ol (tocol) (Matreya Inc., Pleasant Gap, PA, USA) was used as internal standard (IS_1_) for fluorescence analysis and *trans*-apo-8′-carotenal (Fluka, Seelze, Germany) was used as internal standard (IS2) for UV/Vis analysis. All remaining individual standards used for spectrophotometric and chromatographic analyses were of analytical reagent grade and obtained from Sigma-Aldrich (Steinheim am Albuch, Germany). All other reagents and solvents were of analytical or HPLC grade. Ultrapure water (18.2 MW cm resistivity) was used to prepare all the aqueous solutions and was produced using a Simplicity 185 water purification system (Millipore, Molsheim, France).

### 2.2. Material

Chayote leaves were supplied by a local farm located at Cinfães, Douro (Portugal). 1 kg of leaves was collected in October 2021 from 10 plants (random sampling) of green chayote variety at maturity stage to obtain a representative sampling. Leaves of the same color and size were chosen to limit experimental variations and the final quantity of homogeneous leaves was approximately 100 g in the fresh state. Within 24 h from the time of harvest, leaves were examined for integrity and absence of dust and insect contamination, the stems were removed, and the material was cleaned with tap water and dried (Excalibur 9 Tray Dehydrator, Model 4926 T, Armonk, NY, USA) for 12 h at 35 °C. The dry matter (DM) of chayote leaves was 87.6 ± 0.65%. The dried leaves were ground (Moulinex A320), sieved through a 0.75 mm stainless steel sieve, thoroughly mixed and stored at 8 °C under light-free conditions until extractions.

### 2.3. Experimental Design

#### 2.3.1. Selection of Variables

Preliminary studies were undertaken using a univariate method to make sure that possible maximum and minimum points of RSM were achieved. The best extraction solvent was selected from the following seven solvent mixtures: 100% distilled water; 20, 50 and 80% ethanol:distilled water; and 20, 50 and 80% acetone:distilled water. For the selection of the best solvent, 1 g of dried chayote leaf powder was mixed with 20 mL of the solvent and subjected to an ultrasonic bath (Selecta SA Barcelona, Spain), equipped with digital timer, temperature, and sonication power controller. Extractions were conducted at 50 °C, with a treatment time of 20 min, ultrasound amplitude of 60% (170 Watts) and occasional stirring. For determination of best solid to sample ratio, 1 g of sample was mixed with an appropriate volume of the best solvent (10–50 mL), keeping all other experimental parameters constant. The selection of the best condition was based on maximum TPC and TC in the extracts.

#### 2.3.2. BBD Optimization and Validation

Optimization of UAE of phenolics and carotenoids from chayote leaves was assessed using Response Surface Methodology (RSM). The experimental design followed was the Box-Behnken design (BBD) that was prepared using Design Expert Version 7 software (State-Ease Inc., Minneapolis, MN, USA). The BBD comprised three levels and three factors consisting of 17 experiments (Table 2). The three independent variables were time (X1: 30–80 min), temperature (X2: 35–55 °C) and ultrasonic power amplitude (X3: 60–80% US-amplitude). Based on the single-factor experimental design, extractions were performed using 50% ethanol:distilled water and a solid to sample ratio of 1:30 g/mL, with occasional stirring. Each extraction condition was performed in triplicate. Total phenolic content, TPC (Y1), total carotenoid content, TC (Y2) and 2,2′-azino-bis (3-ethylbenzothiazoline-6-sulphonic acid radical scavenging activity, and ABTS-RSA (Y3) were taken as the dependent variables. Desirability indices were constructed to obtain the optimum experimental conditions to maximize the bioactivities of chayote leaves. Confirmatory experiments were performed with the parameters suggested by the experimental model in three different runs, and the *t* test was applied to compare the TPC, TC and ABTS-RSA of chayote leaf extracts prepared under optimized conditions with those predicted by models.

### 2.4. Preparation of Chayote Leaves Extracts

#### 2.4.1. Ultrasound-Assisted Extraction (UAE)

For the UAE procedures, 1 g of dried chayote leaf powder was mixed with 30 mL of the best extraction solvent in a 3.5 cm inner diameter cylindrical flask. After that, the flask was covered with aluminum foil and placed in an ultrasonic bath sonicator (Bandelin SONOREX™ Digital 10 P Ultrasonic baths DK 102 P, Bandelin Electronic GmbH, Berlin, Germany). The extraction was then carried put at different ultrasonic powers and different temperatures for different periods of time, according to the experimental design described in Section 2.3.2.

#### 2.4.2. Maceration Extraction (ME)

ME was performed in the same equipment used for UAE, except without application of ultrasonic power during the extraction process. The experimental conditions were determined according to the condition of the highest yield achieved in the experimental design of UAE.

#### 2.4.3. Microwave-Assisted Extraction (MAE)

MAE was performed with a MARS-X 1500 W (Microwave Accelerated Reaction System for Extraction and Digestion, CEM, Mathews, NC, USA). The experimental conditions were determined according to the condition of the highest yield achieved in the experimental design of UAE and using medium power (300 Watts).

### 2.5. Characterization of Chayote Leaf Extracts

The solutions obtained by UAE, ME and MAE were filtered through Whatman nº 1 paper, centrifuged (Sigma 3-30KS, Sigma, Osterode am Harz, Germany) at 8000 rpm for 10 min and the ethanol was eliminated in the rotary evaporator (Buchi Rotavapor, R-200) at 35 °C. The residue was frozen at −80 °C for subsequent lyophilization (Telstar, model Cryodos-80, Barcelona, Spain). The final extracts were stored at 4 °C and protected from light until analysis. The percent yield of chayote leaf extract was assessed by dividing the weight of the lyophilized extract with the sample weight and multiplying by 100.

#### 2.5.1. Total Phenolic Content (TPC)

The total phenolic content (TPC) was measured spectrophotometrically according to the Folin–Ciocalteu procedure [15], with the modifications by [16]. Briefly, 100 mg of the lyophilized extract was diluted with 10 mL of absolute ethanol. Then, a sample aliquot (25 μL) was mixed with 25 µL Folin–Ciocalteu reagent, 75 µL of distilled water and 100 µL of 75 g/L Na2CO3. The absorbance was measured at 765 nm using a microplate reader (Synergy HT Microplate Reader (BioTek Instruments, Inc., Winooski, VT, USA) after 90 min incubation at 25 °C. The total phenol content was expressed as mg of gallic acid equivalents per g of dry weight of extract (mg GAE/g DW).

#### 2.5.2. Total Carotenoids Content (TC)

Total carotenoids content (TC) in the lyophilized extracts was assessed by the colorimetric method at the wavelengths of 470, 646 and 663 nm, according to [9], with slight modifications. Briefly, 100 mg of lyophilized extract was mixed with 10 mL of 85% acetone in a dark bottle and left at room temperature for 15 h, then filtered into a 50 mL volumetric flask, and made up to volume by 85% acetone solution. A blank experiment, using acetone (85%), was carried out. Contents were expressed in mg carotenoids per g dry weight of extract (mg/g DW).

#### 2.5.3. ABTS Radical Scavenging Activity (ABTS-RSA)

ABTS radical action was performed according to the previously described method in [17]. Briefly, 100 mg of lyophilized extract was diluted with 10 mL of absolute ethanol. Then, a sample aliquot (20 μL) was mixed with 180 μL ABTS solution and incubated in the darkness at 30 °C for 10 min, followed by absorbance reading at 734 nm. The ABTS working solution with an absorbance of 1.1 ± 0.02 at 734 nm was achieved by diluting the stock solution with ethanol. The ABTS scavenging activity was expressed as mg AA equivalents per g dry weight of extract (mg AAE/g DW).

#### 2.5.4. Ferric Reducing Antioxidant Power (FRAP)

A FRAP assay was performed according to the procedure developed by [16] using ascorbic acid (AA) as standard, and the absorbance was measured at 593 nm at 37 °C after 10 min. The results were expressed as mg AA equivalents per g dry weight of extract (mg AAE/g DW).

#### 2.5.5. HPLC Phenolic Composition Profile

The phenolic profile characterization and quantification were performed by HPLC with diode array detection (DAD), as previously described by [18]. Before injection, 50 mg of lyophilized extract was resuspended in 1 mL of methanol:water 50:50 (*v*/*v*) and filtered through a 0.22 μm PTFE filter (MS^®^ nylon membrane filter; Membrane Solutions, MFNY047022). An HPLC system (Shimadzu Corporation, Kyoto, Japan), equipped with a reversed-phase C18 column (250 mm × 4.6 mm, 5 μm, Phenomenex, Torrance, CA, USA) operated at 25 °C, was employed for the phenolic compounds separation. An injection volume of 20 µL, a flow rate of 1.0 mL/min and a mobile phase, in gradient mode, composed by methanol (A) and water (B), both with 0.1% formic acid, were used for sample elution. The identification of phenolic compounds in chayote leaf extracts was carried out by comparing the retention time and UV-vis spectra of detected peaks with those from pure standards. The quantification was performed at three wavelengths (280, 320 and 360 nm), according to the maximum absorption from each compound. Triplicate injections were made, and the results were expressed as mean ± standard deviation (SD).

#### 2.5.6. HPLC Vitamin A, Vitamin E, Carotenoids, and Chlorophylls Composition Profile

Chromatographic analyses were carried out using an integrated system with a data transmitter (Jasco LC–NetII/ADC, Tokyo, Japan), pumps (Jasco PU–4180, Japan), an auto-sampler (Jasco AS–4050, Japan), oven (ECOM Eco2000, Zlin, Czech Republic), a DAD (Jasco MD–4010, Japan), and a dual-channel fluorescence detector (FLD, Jasco FP–4025, Japan). Data were analyzed using the ChromNAV Control Center v2- JASCO Chromatography Data Station. The lyophilized chayote leaf extracts were reconstituted in 1 mL of 1,4-dioxane:n-hexane (1:3 *v*/*v*). Then, IS_1_ (10 µg, in solution), IS_2_ (0.8 µg, in solution) and anhydrous sodium sulfate (100 mg) were added and vortexed for 30 s. Samples were macerated for 30 min with periodic agitation. Subsequently, samples were centrifuged (3 min, 13,000 rpm) and the resulting supernatant was analyzed immediately. The method validation and chromatographic separation adopted was described by [19] and achieved with a normal-phase column (Luna Silica; 100 mm × 3 mm; 3 µm) (Phenomenex, USA) eluted with a total run time of 35 min. gradient from 1,4-dioxane:n-hexane (1:33, *v*/*v*) over 10.5 min, increased to 25% (*v*/*v*) at 17.5 min, which was kept for 1.5 min until it returned to the initial conditions at 20 min. The flow rate was 1 mL/min with the temperature maintained at 22 °C and the injection volume was 20 µL. The compounds were identified by chromatographic comparisons with authentic standards and against UV/Vis spectra comparison. Quantification was based on either UV/VIS (carotenoids and derivatives, 450 nm) or fluorescence signal response [tocopherols (292/328 nm, gain 10) and retinol derivatives (326/496 nm, gain 10)]), using the internal standard method.

### 2.6. Statistical Analysis

Results were presented as mean ± standard deviation (SD) of at least triplicate experiments. Design-Expert software version 7.0 (Stat-Ease Inc., Minneapolis, MN, USA) was used for establishing the experimental design of the optimization process. The adequacy of the model was evaluated using model analysis, coefficient of determination (R^2^) and lack-of-fit test. IBM SPSS Statistics 24.0 software (SPSS Inc., Chicago, IL, USA) was employed to analyze the data from the influence of UAE variables, antioxidant activity (TPC, TC, ABTS, FRAP) and data HPLC analyses. Duncan’s multiple range test, at a significance level of *p* ≤ 0.05, was used for the comparisons of the mean values.

## 3. Results and Discussion

### 3.1. Single-Factor Experimental Analysis

Single-factor experiments were designed to evaluate the influences of solvent extraction composition and solid to solvent ratio on the extraction yields of total phenolic content (TPC) and total carotenoids content (TC) of chayote leaves. The results are presented in Figure 1.

Due to differences in polarity, no solvent is known to be able to extract all bioactive compounds in plants [20]. Therefore, for the determination of optimum solvent composition for the extraction of maximum phenolics and carotenoids from chayote leaves, seven mixtures of solvents with different polarities (100% water, 20, 50 and 80% ethanol: water and 20, 50 and 80% acetone: water) were evaluated. Extractions were performed using a solid solvent ratio of 1:20 g/mL, 60% ultrasound amplitude (170 Watts) for 20 min of extraction, at 50 °C, with occasional stirring. As shown in Figure 1A, the 50% hydro-ethanolic extract showed the highest (*p* < 0.05) TPC (5.10 ± 0.17 mg GAE/g DW), while the hydroalcoholic mixture of 80% ethanol or acetone promoted the highest (*p* < 0.05) extraction of carotenoids from chayote leaves, 0.76 ± 0.09 mg/g DW and 0.88 ± 0.08 mg/g DW, respectively. The TPC results were in accordance with previous studies [21,22,23], which reported that a binary solvent system, such as a 50% ethanol/water mixture, had higher efficiency in the extraction of phenolic compounds, compared to a mono-component solvent system (pure water). For instance, Singh et al. [21] reported that the use of 50% ethanol/water mixture resulted in higher TPC yields from several Cucurbitacea fruits, in comparison to water or other ratios of the ethanol/water mixture. Like the chayote fruit matrix, the mixture of water and ethanol seems to have a synergistic effect, which facilitates the extraction of phenolic compounds from chayote leaves. As seen in Figure 1A, the water extracts presented comparable yields of TPC (4.05 ± 0.23 mg GAE/g DW) but, as expected, the lowest yields of TC (0.03 ± 0.00 mg/g DW). The study by Chang et al. [4] supported the present observation in which the water extracts of chayote shoots had the highest TPC (5.75 ± 0.44 mg GAE/g DW), while the methanol extract had the highest β-carotene content (0.16 ± 0.01 mg/g DW). Moreover, the TC results suggested that the mixture of polar solvents selectively extracted the polar carotenoids (e.g., xanthophylls). Based on these results, a 50% hydro–ethanolic mixture was selected as a suitable solvent to simultaneously extract the phenolics and carotenoid compounds from chayote leaves.

To maximize the extraction efficiencies of phenolics and carotenoids from chayote leaves, four different solid–solvent (50:50 ethanol:water) ratios were evaluated (1:10, 1:20, 1:30 and 1:40), keeping the fixed parameters of 50 °C, 20 min of extraction, 60% ultrasound amplitude, and occasional stirring. As observed in Figure 1B, as the solid–solvent ratio for extraction increased, the TPC and TC values increased. The yields of TC and TPC were significantly (*p* < 0.05) higher at 1:30 g/mL (0.71 ± 0.10 mg/g DW and 5.07 ± 0.13 mg GAE/g DW, respectively) than those observed using a solid–solvent ratio of 1:10 g/mL, which were 0.37 ± 0.03 mg/g DW and 4.11 ± 0.31 mg GAE/g DW, respectively). This pattern was related to the mass transfer principle. However, there was not much difference in the TC and TPC yields between 1:30 and 1:40 solid–solvent ratios. Therefore, for the optimization of the simultaneous extraction of bioactive compounds from chayote leaves, 1:30 solid–solvent ratio was adopted. A solid–solvent ratio of 1:30 mg/L was also reported in the UAE of bioactive compounds from other plant materials, namely kiwi [11], clove [14] and *Plectranthus amboinicus* leaves [24].

### 3.2. Analysis of Response Surface Methodology

#### 3.2.1. Model Fitting

Table 2 shows the experimental extraction conditions and the experimental and predicted values of TC, TPC and ABTS-RSA of chayote leaf extracts. For all the responses, there was a close agreement between the experimental values and the theoretical values predicted by BBD. TC ranged from 0.41 mg/g DW to 0.87 mg/g DW. TPC varied between 3.28 and 5.60 mg GAE/g DW, and ABTS-RSA from 2.10 to 4.12 mg AAE/g DW. The lowest values of the three responses were recorded at 35 °C for a longer extraction period (55 min), and using 100% ultrasound amplitude (280 Watts), while the highest values of TC, TPC and ABTS were observed when extraction occurred at higher temperature (55 °C) for a shorter extraction period (30 min), and using 80% ultrasound amplitude (224 Watts).

The model summary and the results obtained from ANOVA in the response surface quadratic model are shown in Table 3. The adequacy and significance of the 2FI model were evaluated from the analysis of variance through Fisher’s *F* test. The model’s *F* value for TC, TPC and ABTS responses was, respectively, 4.09, 7.33 and 3.26, indicating the high significance of the model. The variable temperature (X2) and the interactive effect of time and temperature variables (X1.X2) had a significant effect on the three responses, while the variable time (X1) and the second-order quadratic effect of time (X1^2^) had a significant effect on the TPC response. The coefficient of determination (R^2^) for checking the fitness of the model was close to 1, indicating that the models explained, respectively, 73.7, 78.1 and 76.2% of the variation in the UAE conditions on the TC, TPC and ABTS-RSA of chayote leaves. The ‘Adeq Precision’ was higher than 4 for the three responses, indicating an adequate signal-to-noise ratio. Moreover, the statistical analysis of variance also revealed a non-significant (*p* > 0.05) lack-of-fit, which further validated the model. Hence, all the quadratic polynomial models in this study were accurate and reliable to predict the TC, TPC and ABTS responses.

#### 3.2.2. Analyses of Response Surfaces

The curve analysis of response surfaces for experimental design is shown in Figure 2, which allowed the prediction of the responses TC (Y1), TPC (Y2) and ABTS (Y3) function of the effects of the time (X1), temperature (X2) and ultrasound power (X3) extraction parameters. Model equations were visualized in the form of three-dimensional surface plots, which were constructed by plotting the response on the *Z*-axis against any two independent variables, while maintaining other variables at their optimal levels.

The response surface 3D plots shown in Figure 2D–F describe the relationship between TPC and the three extraction parameters. The results showed that the lowest TPC (3.28 mg GAE/g DW) was achieved when a lower extraction temperature (35 °C) and longer sonication time (55 min) were applied (run 2 extraction conditions, Table 2). However, TPC increased when the extraction temperature rose to 55 °C (maximum extraction temperature), reaching the value of 5.6 mg GAE/g DW (run 9 extraction conditions, Table 1). This pattern suggested that the increase of extraction temperature could have contributed to the enhanced diffusivity of the solvent into cells, and a higher rate of cavitation bubble formation, and so increased the solubility and desorption of the phenolic compounds from the cells, with consequent enhancement of the TPC value. A similar behavior was observed for the TC (Figure 2A) and antioxidant activity (Figure 2G) responses, where reduced sonication time (30 min) at high extraction temperatures (55 °C) produced chayote leaf extracts with higher values of TC (0.87 mg/g DW) and ABTS-RSA (4.12 mg GAE/g DW), run 9 extraction conditions (Table 1). These observations were in line with other works [24], which also applied the UAE technique to recover phenolic compounds from plant leaves.

The response surface 3D plots shown in Figure 2C,F,I describe the effect of ultrasound power on the bioactivities of the chayote leaf extracts. The TPC value increased by around 70% (from 3.28 to 5.6 mg GAE/g DW, *p* < 0.05) when the ultrasound power increased from 60% (170 Watts) to 80% (224 Watts), corresponding to runs 2 and 9, respectively, as described in Table 2. As suggested by several authors [12,25], this effect could be attributed to improved cavitation and mechanical ultrasound effects, which enabled an increase in the surface contact area between the solid and the liquid, thus causing a higher penetration of the solvent into the plant matrix. From Figure 2F,I, we can see that the ultrasonic amplitude of 80% (224 Watts) also improved the UAE of total carotenoids and antioxidant capacity (ABTS-RSA) from chayote leaves. This observation was in line with other works [11,13], which referred to an amplitude range between 60% and 85% being considered ideal for the yield efficiency of UAE.

#### 3.2.3. Validation of the BBD Model

The optimal UAE conditions to maximize the carotenoid, phenolic, and antioxidant composition of chayote leaves, applying the lowest ultrasound power, were predicted using RSM. For this purpose, the individual desirability of the three responses were combined into a single number and then the greatest overall desirability was searched for. The optimum conditions predicted by the BBD model were temperature of 55 °C, extraction time of 30 min and 60% (170 Watts) of ultrasound power. With a desirability of 92.8%, the predicted responses by the BBD model were: TC of 0.85 mg/g DW, TPC of 5.09 m GAE/g DW and ABTS-RSA of 4.12 mg AAE/g DW. The experimental values agreed within a 95% confidence interval with the predicted values for the three responses: TC of 0.88 ± 0.02 mg/g DW (*p* = 0.104), TPC of 5.38 ± 0.23 mg GAE/g DW (*p* = 0.215), and ABTS-RSA of 4.23 ± 0.13 mg AAE/g DW (*p* = 0.356). Therefore, the adequacy of the model in predicting the optimum UAE conditions of chayote leaves was confirmed.

### 3.3. Comparison between UAE, ME, and MAE

To validate the effectiveness of the optimized UAE methodology on the extraction of carotenoids, phenolics and antioxidant compounds from chayote leaves, a comparison was carried out between UAE, ME, and MAE techniques. ME and MAE extracts were prepared applying the optimal UAE conditions of the present study: 55 °C, 30 min and solid to solvent (50:50 ethanol:water) ratio of 1:30 g/mL. As already mentioned, ME has been the common technique for the extraction of polyphenol and carotenoids from chayote leaves (Table 1), while MAE was applied to this plant material for the first time in the present study. The comparative analysis of in vitro TC, TPC and antioxidant activity of chayote leaf extracts obtained from the three extraction techniques are reported in Figure 3. The comparative composition profile of phenolics and carotenoids were also evaluated and respectively reported in Table 4 and Figure 4.

#### 3.3.1. TC, TPC and Antioxidant Activity

Applying the optimal UAE extraction conditions of 55 °C, 30 min and solid–solvent (50:50 ethanol:water) ratio of 1:30 g/mL, the UAE, ME, and MAE chayote leave extracts presented respective percent yields of 11.8 ± 1.32%, 7.18 ± 1.02% and 9.01 ± 1.70%. The three extracts were characterized regarding the total carotenoid and total phenolic contents, as well as antioxidant capacity, evaluated by ABTS-RSA and FRAP. As can be seen in Figure 3, compared to ME, the optimized UAE technique enabled the obtaining of chayote leaf extracts with higher carotenoids content (0.85 ± 0.02 vs. 0.61± 0.01 mg/g DW), and total phenolics content (5.38 ± 0.28 vs. 4.24 ± 0.41 mg GAE/g DW), as well as higher antioxidant activity, with ABTS-RSA values of 4.23 ± 0.16 vs. 3.54 ± 0.03 mg GAE/g DW and FRAP values of 5.25 ± 0.14 vs. 4.01 ± 0.21 mg GAE/g DW, respectively. In other words, the UAE technique improved the extraction yield of carotenoids and phenolic compounds by ~30% and ~20%, respectively, compared with the maceration technique. The higher efficiency of UAE against the conventional technique was also observed for other kinds of leaf material. For instance, in the study performed by Ji-Min (2021) [11], the structure of kiwi leaves’ surface morphology was examined after extraction, and it was reported that ultrasound treatment produced cell destruction, while maceration only resulted in slightly ruptured cell pores, which could explain its low extraction efficiency. Pudziuvelyte et al. [26] reported that UAE significantly increased the extracted phenolic yield (0.855 mg GAE/g DW) of *lsholtzia ciliata* leaves compared to the maceration method (0.141 mg GAE/g DW); the UAE treatment for 11 min increased the mass fraction of total phenols by 20% compared to maceration extraction for 30 min with 70% ethanol [26].

When compared to MAE, the ultrasound treatment under the optimized conditions also enabled higher (*p* < 0.05) extraction yields of phenolic compounds and antioxidant compounds, but a similar total carotenoids content was obtained. However, it is important to mention that the employment of MAE implies more expensive equipment in comparison to the requirements of an ultrasound bath for UAE. This is the first paper comparing phenolic and carotenoid composition, and related antioxidant properties, of chayote leaves prepared by UAE, ME, and MAE. The results achieved in this work were compared with those found in literature (Table 1). For instance, The TPC values obtained in this work for the leaf extracts prepared by the maceration technique were in the same order as that reported by Chang et al. [4] for the methanolic (5.16 ± 0.09 mg GAE/g DW) and water (5.75 ± 0.44 mg GAE/g DW) extracts of chayote shoots (including leaves, tendrils, and stem). However, the authors’ applied extraction conditions were very different from the present work, which were room temperature, 2 h of extraction time and a solid–solvent ratio of 5:100 g/mL. Compared to the study conducted by Chao et al. [4], the methanolic extract of chayote leaves prepared by maceration at 75 °C and using a solid–solvent ratio of 1:10 g/mL, presented a TPC value 2 times lower (2.62 ± 0.52 mg GAE/g DW) than samples used in this work (4.24 ± 0.41 mg GAE/g DW). By contrast, the TPC value found in the present work was 20 times lower than that reported by Loizzo et al. [8], who reported a value of 89.3 ± 2.3 mg CAE/g DW. However, their results were expressed in chlorogenic acid equivalents (CAE), instead of Gallic acid equivalents (GAE), and no information was given regarding the temperature and time extraction conditions employed. Concerning the application of UAE, only one study reported this technique (Table 1), demonstrating the novelty of this study. The ultrasonic treatment of chayote leaves with 70% ethanol performed by Kim et al. [14] enabled a similar extraction yield (10.7%), but a TPC value 6 times higher (26.5 mg GAE/g DW) than the value found in the present work. Again, no information was given regarding the temperature and time extraction conditions employed, which could justify the differences observed.

#### 3.3.2. Phenolic Composition Profile

HPLC-DAD was employed to evaluate the phenolic composition profile of chayote leaf extracts prepared by the UAE, ME, and MAE techniques. Table 4 summarizes the identified phenolic compounds by chromatographic analysis, which could contribute to the antioxidant activity observed in the three extracts.

The UAE extract presented the highest (*p* < 0.05) sum of all the identified phenolic compounds, 2.34 ± 0.12 mg/g DW, followed by MAE (2.21 ± 0.12 mg/g DW) and ME (2.04 ± 0.14 mg/g DW) extracts. These results confirmed the highest efficiency of the ultrasound technique in extracting phenolic compounds from chayote leaves, compared to maceration and microwave extraction approaches. Furthermore, the results of the chromatographic analysis were strongly correlated (*R* = 0.8576) with the results of the TPC assay.

According to Table 4, phenolic acids and flavonols were the principal constituents of the chayote leaf extracts. The UAE extract presented the highest content of flavonols (55%), while the phenolic acid fraction was significantly higher (*p* < 0.05) in the MAE extract, being around 33%. Instead, the contribution of flavanols, flavanones and resveratrol were significantly higher in ME extracts, with respective mean values of 9.59%, 5.41% and 2.77%. Ferulic, cinnamic and chlorogenic were the predominant phenolic acids found in all chayote leaf extracts, with mean contents ranging, respectively, from 16.70 mg/100 g DW (ME extract) to 23.73 mg/100 g DW (UAE extract), from 16.70 (ME extract) to 23.73 mg/100 g DW (UAE extract) and from 16.70 (ME extract) to 23.73 mg/100 g DW (UAE extract). The MAE extract presented the highest mean content of gallic acid (1.97 ± 0.67 mg/100 g DW), but the lowest content of protocatechuic acid (0.61 ± 0.02 mg/100 g DW). The compounds 4-hydroxyphenilacetic acid, vanillic acid and syringic acid were not detected in any of the chayote leaf extracts. Catechin and epicatechin were identified and quantified in all chayote leaf extracts, with UAE extracts presenting significantly higher (*p* < 0.05) amounts, 18.02 ± 0.90 and 3.93 ± 0.20 mg/100 g DW, in comparison to MAE (15.02 ± 1.07 and 3.01 ± 0.22 mg/100 g DW) and ME (16.99 ± 1.90 and 2.56 ± 0.17 mg/100 g DW) extracts. The flavanones naringin, naringenin and pinocenbrin were also identified and quantified in the three extracts; naringin and naringenin were found in higher amounts in the ME extract, with respective values of 6.79 ± 0.42 and 2.26 mg/100 g DW. Regarding the flavonols class, myricetin was the major compound found in the three extracts, with respective mean contents of 94.93 ± 4.75, 81.93 ± 5.07 and 84.96 ± 4.01 mg/100 g DW in UAE, ME, and MAE extracts. Kampferol and its derivatives (kaempferol-3-O-glucoside and kaempferol-3-O-rutinoside) were also quantified in representative amounts in the chayote leaf extracts, with the UAE extract exhibiting the highest amounts (5.33 ± 0.27 mg/100 g DW). The levels of rutin, quercetin and tiliroside were in the same range in the three extracts. Resveratrol was also found in all chayote leaf extracts, with the ME extract exhibiting the highest result (5.65 ± 0.17 mg/100 g DW).

To the best of our knowledge, this is the first study that compares the phenolic profile of chayote leaf extracts prepared by ultrasound, microwave, and maceration extraction techniques. The literature contains only two studies which have focused on the phenolic composition of chayote leaf extracts [7,27], and both adopted the maceration extraction technique. Siciliano et al. [6] prepared chayote leaves extracts by exhaustive maceration with chloroform and methanol, at room temperature for 48 h, and characterized eight flavonoids by liquid chromatography-mass spectrometry. The authors reported a total amount of flavonoids of 3.50 mg/g DW, represented by apigenin 6-*C*-ß-*D*-glucopyranosyl-8-*C*-ß-*D*-apiofuranoside, vitexin, luteolin 7-*O*-rutinoside, luteolin 7 *O*-ß-*D*-glucopyranoside, apigenin 7-*O*-rutinoside and diosmetin 7-*O*-rutinoside [6]. In the present study, these compounds were not present in the standard polyphenol mixture available but a further HPLC analysis with mass spectrometry detection should be performed to identify these compounds. Chao et al. [7] compared the phenolic profile of methanolic leaf extracts obtained from green and yellow chayote varieties. According to the chromatographic analysis, myricetin was the principal compound in green and yellow varieties (75.61 ± 4.99 mg/100 g DW and 101.05 ± 3.10 mg/100 g DW; respectively), followed by morin (19.50 ± 0.69 mg/100 g DW and 40.43 ± 8.23 mg/100 g DW), Table 1. The flavonols, quercetin and kaempferol, were only detected in the yellow variety, with mean contents of 6.48 ± 0.28 µg/g DW and 3.63 ± 0.58 µg/g DW, respectively [7]. Myricetin and kaempferol recorded similar findings with the present study, but the quercetin levels of UAE, ME, and UAE extracts were 2 times lower than those found by Chao et al. [7]. Overall, the presence of these compounds could explain the antioxidant activity evidenced by the chayote leaf extracts. The therapeutic role of catechin, resveratrol, quercetin, myricetin, kaempferol, and their glycosylated forms, as antioxidant, anti-inflammatory, anti-cancer, or even immunomodulatory agents for human health is well documented in the literature [6,27,28].

#### 3.3.3. Vitamin A, Vitamin E, Carotenoid, and Chlorophyll Composition Profile

Figure 4 shows the vitamin A, vitamin E, carotenoid, and chlorophyll composition of chayote lea extracts prepared by UAE, ME, and MAE techniques. Based on their spectrum properties, compounds were grouped in different classes: vitamin E precursors (tocopherol esters, α-tocopherol and other tocols), vitamin A precursors (retinol esters, β-carotene equivalents), other carotenoids and xanthophylls. For all the extracts, the sum of these classes of compounds, identified by the HPLC-DAD/FLD analysis, agreed with total carotenoid values from the spectrophotometric assay and confirmed the highest efficiency of UAE in the extraction of these valuable compounds. The sum of the identified classes followed this trend: UAE extract (1.10 ± 0.05 mg/g DW) > MAE extract (0.85 ± 0.02 mg/g DW) > ME extract (0.80 ± 0.06 mg/g DW). Thus, like the results reported for the phenolic compounds, the use of ultrasound significantly improved the efficiency of the extraction process of vitamin A, vitamin E, carotenoids, and chlorophylls from chayote leaves.

As depicted in Figure 4, xanthophylls were the predominant class of carotenoid found in the chayote leaf extracts, corresponding to 50% of the carotenoid composition. The contents of xanthophylls varied between extracts, being significantly higher (*p* < 0.05) in the UAE extract (0.61 mg/g DW), followed by MAE (0.45 mg/g DW) and ME extracts (0.32 mg/g DW). Additional HPLC analysis, with mass spectrometry detection, should be performed to identify the main xanthophyll present in the chayote extracts and to understand their potential bioactive properties. The second major class of carotenoids present in the chayote leaf extracts were β-carotene equivalents and the concentration found ranged between 0.16 (MAE extract) and 0.20 mg/g DW (UAE and ME extracts). These results suggested that chayote leaves present lower levels of β-carotene equivalents than spinach (0.69 mg/g DW), cauliflower (0.55 mg/g DW), carrot (0.48 mg/g DW) and pumpkin (0.48 mg/g DW) [21,29]. To our best knowledge, only Sriwichai et al. [5] characterized the carotenoid profile of chayote leaves, reporting lutein and β-carotene contents of 7.4 and 4.4 mg/100 g FW, respectively.

The main class representing the vitamin E precursors of chayote leaves was α-tocopherol, and the mean contents were similar among UAE and ME extracts, around 0.11 mg/g DW. This content found for the chayote leaves was higher than that reported for spinach leaves, 0.075–0.088 mg/g DW [30], suggesting that chayote leaves might have the potential to supply nutritionally relevant vitamin E in the diet.

## 4. Conclusions

This study successfully applied RSM as a practical approach to optimize the UAE conditions of phenolics and carotenoids from chayote leaves. The second-order polynomial model with high correlation provided adequate mathematical descriptions and effectively optimized the extraction conditions of polyphenolic and carotenoid compounds from this plant material. Under the optimized conditions, 55 °C, 30 min, 60% (170 Watts) ultrasound power, and a solid–solvent (50:50 ethanol:water) ratio of 1:30 g/mL, the developed UAE process showed more efficiency than maceration and microwave-assisted extraction processes in the extraction of polyphenols (5.38 ± 0.28 mg GAE/g DW) and carotenoids (0.85 ± 0.02 mg/g DW). The phytochemical profile of the UAE extract using HPLC-DAD identified the presence of various phenolic compounds, such as myricetin, kaempferol-3-*O*-rutinoside, rutin, quercetin, (+)-catechin, naringin and chlorogenic, ferulic and cinnamic acids, as well as different classes of carotenoids, mostly represented by xanthophylls and β-carotene equivalents, and α-tocopherol, which could explain the antioxidant capacity evidenced by the extracts. To conclude, the results from this study showed that chayote leaves could be a potential source of natural phytochemicals, and UAE offers a cost effective and time efficient approach for the extraction of added-value compounds from chayote leaves, with promising potential as antioxidants in food, and in the pharmaceutical and cosmetic fields.

## Figures and Tables

**Figure 1 molecules-27-07193-f001:**
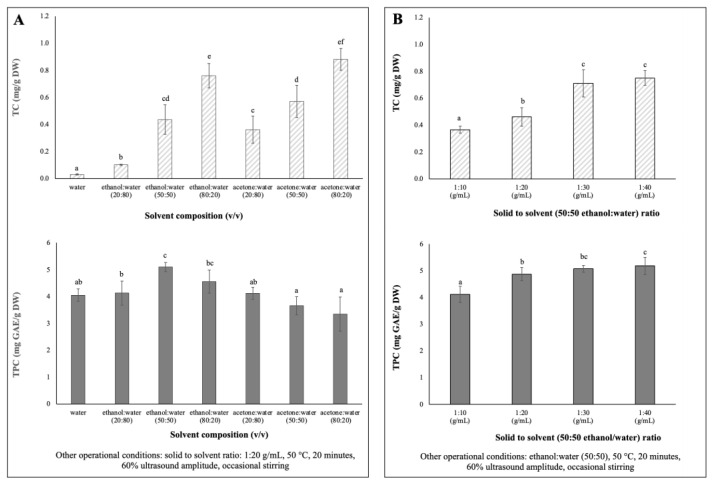
Single factor experiments of the UAE parameters’ effects on TC (mg/g DW) and TPC (mg/g DW) extracted from chayote leaves: (**A**) solvent composition: water, ethanol, acetone (**B**) solid to solvent ratio (g/mL). Results were expressed as mean ± standard deviation (n = 3). Different letters on top of bars (a–f) in the same group indicate significant differences (*p* < 0.05) between means according to Duncan’s multiple range test.

**Figure 2 molecules-27-07193-f002:**
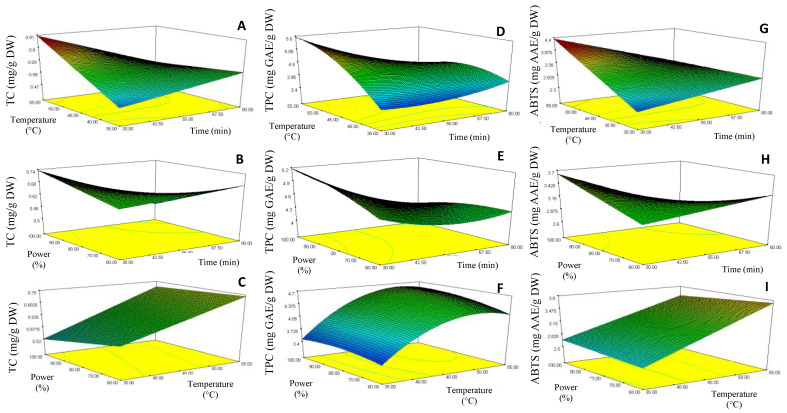
Response surface contour plots showing the combined effects of the extraction parameters (coded values) on the Total carotenoids, TC (**A**–**C**), Total phenolic content, TPC (**D**,**E**) and ABTS-RSA (**G**–**I**) of chayote leaves UAE extracts.

**Figure 3 molecules-27-07193-f003:**
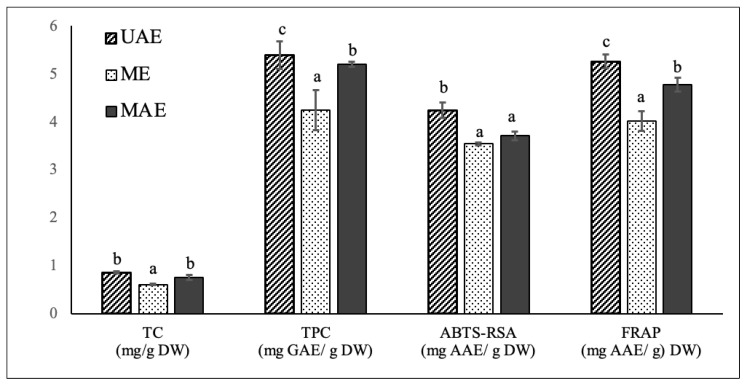
Comparison of total carotenoids content (mg/g DW), total phenolic content (mg GAE/g DW) and antioxidant capacity, evaluated by ABTS-RSA and FRAP assays, of chayote leaves extracts obtained by ultrasound-assisted extraction (UAE), maceration extraction (ME) and microwave assisted extraction (MAE) techniques. The results were expressed as mean ± standard deviation (n = 3). Different letters on top of bars (a–c) in the same group indicate significant differences (*p* < 0.05) between means according to Duncan’s multiple range test.

**Figure 4 molecules-27-07193-f004:**
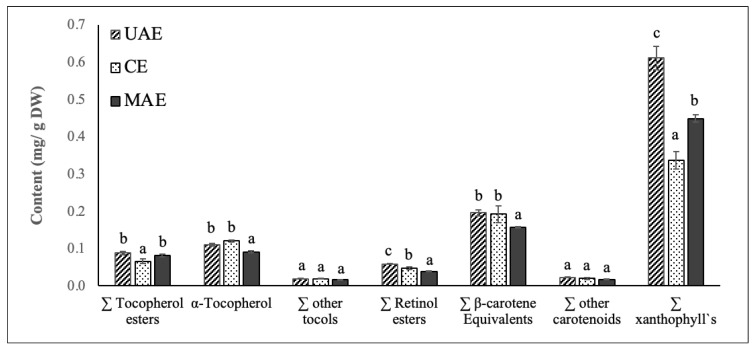
Content (mg/g DW) of vitamin E precursors (tocopherol esters, α-tocopherol and other tocols), vitamin A precursors (retinol esters, β-carotene equivalents), other carotenoids and xanthophyll`s identified in chayote leaves extracts obtained by UAE, ME, and MAE. Results were expressed as mean ± standard deviation (n = 3). Different letters on top of bars (a–c) in the same group indicate significant differences (*p* < 0.05) between means according to Duncan’s multiple range test.

**Table 1 molecules-27-07193-t001:** Summary of the studies that have evaluated the yield and composition of chayote leaves extracts.

Extraction Method	Extraction Parameters	Extraction Yield (%)	TPC (/g DW of Extract)	Carotenoids (/g DW of Extract)	Antioxidant Activity (/g DW of Extract)	Reference
Ultrasound (leaves)	Ethanol/water (50:50) 1:30 g/mL 30 min, 55 °C, 224 W	11.8 ± 1.32%	5.38 ± 0.28 mg GAE/g DW	TC: 0.85 ± 0.02 mg/g DW ∑ β-carotene equivalents 0.20 ± 0.01 mg/g DW	ABTS: 4.23 mg AAE/g DW FRAP: 5.25 mg AAE/g DW	Present study
Ultrasound (leaves)	70% Ethanol	10.7%	26.5 mg GAE/g DW	ND		[9]
Microwave (leaves)	Ethanol/water (50:50) 1:30 g/mL 30 min, 55 °C, 300 W	9.01 ± 1.70%	5.20 ± 0.05 mg GAE/g DW	TC: 0.76 ± 0.05 mg/g DW ∑ β-carotene equivalents 0.16 ± 0.01 mg/g DW	ABTS: 3.71 mg AAE/g DW FRAP: 4.77 mg AAE/g DW	Present study
Maceration (leaves)	Ethanol/water (50:50) 1:30 g/mL 30 min, 55 °C	7.18 ± 1.02%	4.24 ± 0.41 mg GAE/g DW	TC: 0.61 ± 0.01 mg/g DW ∑ β-carotene equivalents 0.19 ± 0.02 mg/g DW	ABTS: 3.54 mg AAE/g DW FRAP: 4.01 mg AAE/g DW	Present study
Maceration (shoots-leaves, tendrils and stem)	hexane ethyl acetate methanol water 5:100 g/mL 2 h, room temperature	0.67% (hexane)– 24.04% (water)	TPC, mg GAE/g DW: Hexane: 0.14 ± 0.02 Ethyl acetate: 0.68 ± 0.02 Methanol: 5.16 ± 0.09 Water: 5.75 ± 0.44	β-carotene content, mg/g DW: Hexane: 0.06 ± 0.01 Ethyl acetate: 0.07 ± 0.01 Methanol: 0.16 ± 0.01 Water: ND		[4]
Maceration (leaves)	Methanol	2.5%	89.3 ± 2.3 mg CAE/g DW	TC: 0.05 ± 0.01 mg/g DW	FRAP: 1.24 mg Fe(II)/g DW	[8]
Maceration (leaves)	Methanol (1.2 N HCl), 75 °C, 1:10 g/mL	NI	Green variety: 2.62 ± 0.52 mg GAE/g DW Myricetin: 75.61 ± 4.99 mg/100 g DW; Quercetin: ND; Kaempferol: ND; Morin: 19.50 ± 0.69 mg/100 g DW Yellow variety: 0.63 ± 0.18 mg GAE/g DW Myricetin: 101.05 ± 3.10 mg/100 g DW; Quercetin: 6.48 ± 0.28 mg/100 g DW; Kaempferol: 3.64 ± 0.58 mg/100 g DW; Morin: 40.44 ± 8.23 mg/100 g DW	ND	Green variety: ABTS (ethanolic extract): 0.58 ± 0.07 mg TE/g DW Yellow variety: ABTS (ethanolic extract): 0.32 mg ± 0.06 TE/g DW	[7]

Legend: AAE, ascorbic acid equivalents; CAE, chlorogenic acid equivalents, GAE gallic acid equivalents; TE, Trolox equivalents; Ferric reducing ability power (FRAP), Antioxidant capacity determined by radical cation (ABTS), TC, total carotenoids. NI, not informed; ND; not detected.

**Table 2 molecules-27-07193-t002:** Experimental conditions and results of total carotenoids (Y1), total phenolics content (Y2) and antioxidant activity (Y3) obtained by UAE of chayote leaves.

Independent Variables				Investigated Responses					
Run	X1 Time (min)	X2 Temperature (°C)	X3 Power (%)	Y1-TC (mg/g DW)		Y2-TPC (mg GAE/g DW)		Y3-ABTS (mg AAE/g DW)	
				Exp.^a^	Pred.^b^	Exp.^a^	Pred.^b^	Exp.^a^	Pred.^b^
1	55	55	60	0.71	0.74	4.21	4.25	3.54	3.74
2	55	35	60	0.51	0.59	3.28	3.43	2.14	2.59
3	80	35	80	0.67	0.63	3.90	3.95	3.18	2.97
4	55	55	100	0.66	0.72	4.70	4.56	3.22	3.55
5	55	45	80	0.67	0.64	4.56	4.43	3.33	3.14
6	80	45	100	0.56	0.51	4.09	4.08	2.78	2.60
7	55	45	80	0.59	0.64	4.00	4.43	3.01	3.14
8	30	45	100	0.81	0.74	4.99	5.18	4.01	3.63
9	30	55	80	0.87	0.90	5.60	5.55	4.12	4.35
10	55	35	100	0.41	0.52	3.50	3.46	2.10	2.68
11	55	45	80	0.57	0.64	4.27	4.43	2.67	3.14
12	30	35	80	0.49	0.48	3.67	3.52	2.32	2.30
13	80	45	60	0.71	0.69	4.55	4.36	3.34	3.30
14	30	45	60	0.69	0.65	4.55	4.56	3.26	3.02
15	55	45	80	0.77	0.64	4.78	4.43	3.77	3.14
16	80	55	80	0.55	0.56	3.67	3.82	2.91	2.94
17	55	45	80	0.72	0.29	4.56	4.43	3.67	3.14

^a^ Experimented values are expressed as average of triplicate determinations from different experiments. ^b^ Predicted valued based on BBD evaluation.

**Table 3 molecules-27-07193-t003:** Model summary and analysis of variance (ANOVA) for TC (mg/g DW), TPC (mg GAE/g DW), and ABTS-RSA (mg AAE/g DW) of chayote leaves in response surface quadratic model.

Source	Mean Square			F Value				*p* Value	
	Y1	Y2	Y3	Y1	Y2	Y3	Y1	Y2	Y3
Model	0.028	0.562	0.642	4.09	7.33	3.26	0.025 *	0.008 *	0.048 *
X1-Time (min)	0.017	0.845	0.281	2.54	11,03	1.43	0.142	0.013 *	0.259
X2-T (°C)	0.063	1.837	2.040	9.34	23.99	10.38	0.013 *	0.002 **	0.009 **
X3-Power (%)	0.004	0.059	0.004	0.60	0.77	0.02	0.456	0.410	0.889
X1.X2	0.063	1.166	1.082	9.27	15.23	5.50	0.012 *	0.006 **	0.041 *
X1.X3	0.018	0.203	0.423	2.70	2.64	2.15	0.131	0.148	0.173
X2.X3	0.001	0.018	0.020	0.09	0.23	0.10	0.767	0.646	0.759
X12		0.166			2.17			0.184	
X22		0.752			9.82			0.017 *	
X32		0.032			0.42			0.536	
Residual	0.007	0.077	0.197						
Lack-of-fit	0.007	0.056	0.188	0.92	0.62	0.90	0.561	0.640	0.571
Pure error	0.007	0.092	0.210						
R^2^ pred (Y1)-0.9105; R^2^ adjust (Y1)-0.7367; Adeq. Precision (Y1)-8.11 R^2^ pred (Y2)-0.9041; R^2^ adjust (Y2)-0.7808; Adeq. Precision (Y2)-10.03 R^2^ pred (Y3)-0.8591; R^2^ adjust (Y3)-0.7619; Adeq. Precision (Y3)-7.21									

Y1, TC (mg/g DW); Y2, TPC (mg GAE/g DW); Y3, ABTS-RSA (mg AAE/g DW). * Significance at *p* < 0.05; ** significance at *p* < 0.01.

**Table 4 molecules-27-07193-t004:** Content (mg/100 g DW) of the identified phenolic compounds in chayote leaf extracts prepared by UAE, ME, and MAE. The results were expressed as mean ± standard deviation (n = 3).

Compounds	UAE (mg/100 g DW)	ME (mg/100 g DW)	MAE (mg/100 g DW)
Gallic acid	1.01 ± 0.05	0.77 ± 0.15	1.97 ± 0.67
Protocatechuic acid	0.81 ± 0.04	0.84 ± 0.06	0.61 ± 0.02
4-hydroxyphenilacetic acid	ND	ND	ND
4-hydroxybenzoic acid	1.92 ± 0.10	1.23 ± 0.96	1.97 ± 0.09
4-hydroxybenzaldehyde	2.51 ± 0.13	2.45 ± 0.15	2.87 ± 0.09
Chlorogenic acid	21.25 ± 1.06	15.25 ± 1.00	22.52 ± 0.87
Vanillic acid	ND	ND	ND
Caffeic acid	2.69 ± 0.13	2.55 ± 0.21	2.85 ± 0.11
Syringic acid	ND	ND	ND
*p*-coumaric acid	3.12 ± 0.16	4.99 ± 0.19	5.13 ± 0.22
Ferulic acid	23.73 ± 1.19	16.70 ± 1.15	22.73 ± 1.08
Sinapic acid	3.12 ± 0.16	3.10 ± 0.13	3.12 ± 0.12
Cinnamic acid	5.86 ± 0.29	5.59 ± 0.23	8.86 ± 0.32
∑ Phenolic acids	66.02 ± 3.30	53.47 ± 4.25	72.63 ± 3.60
(+)-Catechin	18.02 ± 0.90	16.99 ± 1.90	15.02 ± 1.07
(-)Epicatechin	3.93 ± 0.20	2.56 ± 0.17	3.01 ± 0.22
∑ Flavanols	21.95 ± 1.10	19.55 ± 2.07	18.03 ± 1.29
Naringin	5.98 ± 0.30	6.79 ± 0.42	3.66 ± 0.29
Naringenin	1.06 ± 0.05	2.26 ± 0.30	2.13 ± 0.08
Pinocenbrin	2.13 ± 0.11	1.98 ± 0.05	1.15 ± 0.04
∑ Flavanones	9.16 ± 0.46	11.03 ± 0.77	6.93 ± 0.41
Rutin	5.73 ± 0.29	5.45 ± 0.25	5.73 ± 0.66
Quercetin-3-O-glucopyranoside	ND	ND	ND
Quercetin-3-O-galactoside	1.41 ± 0.07	0.99 ± 0.03	1.05 ± 0.05
Myricetin	94.93 ± 4.75	81.93 ± 5.07	84.96 ± 4.01
Kaempferol-3-O-glucoside	3.28 ± 0.16	3.84 ± 0.18	3.28 ± 0.22
Kaempferol-3-O-rutinoside	12.01 ± 0.60	9.06 ± 0.42	10.01 ± 0.62
Quercetin	3.98 ± 0.20	3.06 ± 0.35	2.98 ± 0.19
Tiliroside	2.28 ± 0.11	2.05 ± 0.14	2.78 ± 0.14
Kaempferol	5.33 ± 0.27	3.69 ± 0.18	4.23 ± 0.18
∑ Flavonols	128.96 ± 6.45	110.06 ± 6.62	115.04 ± 6.06
∑ Stilbenes (Resveratrol)	4.69 ± 0.23	5.65 ± 0.17	4.69 ± 0.35
Phloridzin	2.55 ± 0.13	2.43 ± 0.23	2.85 ± 0.18
Phloretin	1.11 ± 0.06	1.69 ± 0.15	1.45 ± 0.05
∑ Others	3.66 ± 0.18	4.12 ± 0.37	4.30 ± 0.23
∑ All phenolic compounds	234.45 ± 11.72	203.88 ± 14.25	221.63 ± 11.94

ND: not detected.

## Data Availability

Not applicable.

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
