# Peer review of "Valorization of Phenolic and Carotenoid Compounds of Sechium edule (Jacq. Swartz) Leaves: Comparison between Conventional, Ultrasound- and Microwave-Assisted Extraction Approaches"

_molecules, 2022, doi:10.3390/molecules27217193_

Round 1

Reviewer 1 Report

The paper gives a detailed insight into the extraction possibilities as well as the nutrient composition of the leaves of the species Sechium edule. Considering that the fruits of this species are more widely used for food purposes, the analysis of the leaf is extremely important, especially because the leaf is the most common by-product that often ends up as waste and is underutilized. In this paper, the possibilities of leaf utilization are presented, extraction techniques are studied, and possible solutions for its use are presented.

The introduction is concisely written and clearly and argumentatively explains the themes of the paper.

The materials and methods are described in detail, and minor suggestions for improvement are listed at the bottom of the text.

The results are clearly presented, and all figures and tables are necessary to make them clearer. The discussion is clearly and comprehensively stated. The authors have studied the topic in depth through other relevant scientific research and have supported their findings with current references.

The conclusion clearly and concisely summarizes the major findings of the research.

In view of all this, I propose to the Editorial Board that this paper be approved for publication with a minor revision in accordance with the suggestions listed below.

ABSTARCT

Line 16 - please indicate the type of ultrasound machine used for the UAE.

Line 21-22 - please indicate which extraction technique (ME, MAE or UAE) was the best solution for the extraction of polyohenols and carotenoids

INTRODUCTION

Line 69 - Table 1 - since this is not a review article, the table is not required

Materials and methods

Line 120 - "in processed foods" - it is not clear what type of device was used. Contrary to the indicated model in the parenthesis, it is a deyhdrator, i.e. a convective dryer. 

Line 130 - it is stated that the chayote powder used for the extraction procedures was freeze-dried, but the text does not describe this procedure. 

Line 168 - for MAE it is important to indicate the power used for extraction. 

Line 216 - please indicate the model of the HPLC system.

Line 232 - "lyophilized chayote leaf extracts" - as mentioned above, it is not described that the plant material used for the analysis was lyophilized (freeze-dried).

Author Response

Dear Editor,

I am submitting the revised version of the article entitled: “Valorization of phenolic and carotenoids compounds of Sechium edule (Jacq. Swartz) leaves: Comparison between conventional, Ultrasound- and Microwave-Assisted extraction approaches” with the Ref.: molecules-1975513 for publication in the Molecules special issue 2nd Edition of Food Bioactives: Chemical Challenges and Bio-Opportunities.

The text of the submitted version was changed according to the answers of the two reviewers. The lines that I’m going to refer correspond to the revised marked manuscript (highlighted version).

I hope that these changes allow the publication of the respective manuscript.

Many thanks for your attention.

Sincerely yours,

Elsa F. Vieira

Reviewer 2 Report

The manuscript investigated the three different extraction methods of phytochemicals in Sechium edule leaves. The results showed that ultrasound could be an effective extraction technique to obtain bioactive compounds from chayote leaves. The research is general  satisfying. Therefore, some item should be revised before acceptance for publication.

Title

"Carotenoids" should be changed as "Carotenoid".

Abstract

Line 16, change "carotenoids" to "carotenoid"

Line 19, change "activity" to "activities"

Keywords

combine chayote by-product and leaves to one keyword as "chayote leaves“.

Material and methods

Line 186, "Na2CO3" could be "Na2CO3", the number should be subscript.

Line 233, what the means of "IS", the full name should be provided.

Line 243 and Line 244, which wavelength for carotenoids analysis at UV/VIS condition, and excitation and emission wavelength for tocopherols analysis at fluorescence conditon? And also, I think retinol could be as carotenoid derivatives and detecte at UV/VIS. Please check it.

Line 245, how to use internal standard method for both carotenoids and tocopherols analysis with different dectors? Which internal standard was used in this study? What is the concentrention of internal standard used in this study? How about the recovery of the standard for this methods?

Results and Disscusion

Tocopherols are not belong to carotenoids family, tocopherols belong to tocochromanols family. It should be clarified in the manuscript and discussed as individual paragraph. How about the other tocols? How to calculate them?

Author Response

(The authors gave the same response as above.)

Round 2

Reviewer 2 Report

The manuscript could be accepted as this version.